# Effects of Water Restriction and Water Replenishment on the Content of Body Water with Bioelectrical Impedance among Young Adults in Baoding, China: A Randomized Controlled Trial (RCT)

**DOI:** 10.3390/nu13020553

**Published:** 2021-02-08

**Authors:** Jianfen Zhang, Na Zhang, Songming Du, Shufang Liu, Guansheng Ma

**Affiliations:** 1Department of Nutrition and Food Hygiene, School of Public Health, Peking University, 38 Xue Yuan Road, Haidian District, Beijing 100191, China; zjf@bjmu.edu.cn (J.Z.); ziqingxuanping@126.com (N.Z.); 2Laboratory of Toxicological Research and Risk Assessment for Food Safety, Peking University, 38 Xue Yuan Road, Haidian District, Beijing 100191, China; 3Chinese Nutrition Society, Room 1405, Beijing Broadcasting Building, No. 14 Jianguomenrai Street, Chaoyang District, Beijing 100191, China; songmingdu097@126.com; 4School of Public Health, Hebei University Health Science Center, 342 Yuhua Road, Lianchi District, Baoding 071000, China; shufangliu@126.com

**Keywords:** water restriction, dehydration, water replenishment, rehydration, body composition

## Abstract

Insufficient water intake may affect body composition. The purpose of this research was to explore the effects of water restriction and replenishment on body composition and to evaluate the optimum amount of water that improves body composition. A total of 76 young adults aged 18–23 years old (40 males and 36 females) in Baoding, China, were recruited in this randomized controlled trial, with a 100% completion rate. After fasting overnight for 12 h, at 8:00 a.m. of day 2, a baseline test, including anthropometric indices and collection of urine and blood samples, was explored. Participants were then subjected to water restriction for 24 h, and three meals with ≤75% water content were provided. At 8:00 AM of day 3, the same indices were determined as a dehydration test. Then, participants were randomly assigned into four groups: three water replenishment groups (WR groups 1, 2, and 3 given 1000, 500, and 200 mL of purified water, respectively) and one non-replenishment group (NR group, with no water). After 90 min, the same measurements were performed as a rehydration test. Compared with the baseline test, during the dehydration test, the intracellular water to total body water ratio (ICW/TBW) increased; and extracellular water (ECW), ECW/TBW (extracellular water to total body water ratio), and TBW decreased (all *p* < 0.05). For males, significant differences were found in ECW, ECW/ICW (extracellular water to intracellular water ratio), ICW/TBW, and ECW/TBW (all *p* < 0.05); for females, significant reductions were found in ICW, ECW, TBW, ECW/ICW, ICW/TBW, and ECW/TBW (all *p* < 0.05). Furthermore, significant differences were found in ICW, ECW, ICW/TBW, ECW/TBW, ECW/ICW, TBW, and TBW/BW between males and females during the baseline and dehydration test (all *p* < 0.05). Comparing the dehydration test with the rehydration test, there were significant interactions between time × volume in ICW and TBW (*F* = 3.002, *p* = 0.036; *F* = 2.907, *p* = 0.040); in males, these were only found in ICW (*F =* 3.061, *p* = 0.040); in females, they were found in ICW and TBW (*F* = 3.002, *p* = 0.036; *F* = 2.907, *p* = 0.040). The ICW levels in WR groups 1 and 2 were all higher than in the NR group (all *p* < 0.05); the TBW was higher in WR group 1 than in the NR group (*p* < 0.05). No significant differences were found between WR groups 1 and 2, either in males or in females (all *p* > 0.05). In the rehydration test, significant differences in body composition were found between males and females among the four groups (all *p* < 0.05). Water restriction had adverse effects on body composition, and females were more susceptible to water restriction than males. Water replenishment improved the water content of body composition, alleviating the adverse effects of water restriction on ICW and TBW. After water restriction for 36 h, the optimum volume of water to improve body composition among young male adults was 1000 mL, but this was not the case for females.

## 1. Introduction

Water was of vital importance for the development of life on Earth [1]. Furthermore, water is a biomarker of changes in the cellular environment in live animals [2]. For humans, water is the main component of the body and is essential for cellular homeostasis [3]. As we age, the proportion of water in our bodies decreases, from 75% of our body weight as infants, to 60% in children, and 55% in the elderly [3]. The input and output of water is in a dynamic balance among healthy adults, which is maintained at about 2500 mL/d [4]. The body’s water content consists of intracellular water (ICW) and extracellular water (ECW), which represent about 2/3 and 1/3 of total body water (TBW), respectively [5]. ICW is the main determinant of cell volume, and ECW includes plasma, interstitial fluid, and other transcellular fluids [1]. Therefore, water is essential for every process in the human body.

The water intake of humans is mainly from drinking fluids, water from food, and metabolic water. Approximately 250–350 mL/day of water is from the metabolism of protein, fat, and carbohydrates, representing about 5–10% of human water intake [1]. It has been shown that the main fluid regulatory process of total water intake (TWI) in humans is drinking, mediated through the sensation of thirst [4]. Thirst stimulation drives people to increase fluid consumption in order to increase TBW. The output of water is mainly from urine, which is mediated by vasopressin secreted by the posterior pituitary gland, participating in the collecting duct to promote free-water reabsorption at the nephron [1]. In normal conditions, water homeostasis is kept relatively stable through osmotic pressure, due to the osmotic balancing action of solutes in ICW and ECW. Moreover, the osmotic balance of ICW and ECW depends on the diffusion of water inside and outside the cell. Some studies exploring TBW among adults in free-living conditions have demonstrated that TBW varies due to differences in age, gender, and country, with TBW ranging from 38 to 46 L in males and 26 to 33 L in females [6,7].

Studies have shown that body composition is associated with health [8,9,10]. Studies have demonstrated that TBW and ICW are strongly correlated with muscle mass in elderly people, and those with higher ICW had a better functional performance and a lower frailty risk [11]. In cases of healthy people, the extracellular water (ECW) to total body water (TBW) ratio (ECW/TBW) can be maintained at a constant value (0.38). The ECW/TBW may be an indicator for predicting therapeutic durability in advanced lung cancer and is a useful indicator for nutritional assessment [12]. However, an increase in ECW/TBW may reflect locomotive syndrome risk and frailty [13,14], chronic liver diseases [15], renal disorders [16], coronary artery calcification [17], and heart failure [18]. In fact, the extracellular water (ECW) to intracellular water (ICW) ratio (ECW/ICW) was associated with nutritional status [19], and it can be increased with age among healthy adults [20]. In hydration assessment, the ECW/TBW is often used as an indicator of volume overload [21]. In some disease states, including hypoalbuminemia and increased vascular permeability caused by inflammation, ECW increases, while ICW decreases and the ECW/ICW increases [22]. Therefore, it is necessary to maintain the stability of the body’s composition. Moreover, observing the changes in body composition is of vital importance for human health [23].

Studies have largely focused on the associations between TWI and disease, whereas the importance of body composition and hydration status has not had enough attention. Therefore, more related studies need to be initiated. Few studies have confirmed the relationship between fluid intake and body composition. A cross-sectional study conducted among young adults in China revealed a moderate correlation between water intake and body composition (such as the TBW/BW), in both males and females [24]. Similarly, in children, the water intake normalized by body weight is positively correlated with body water content, for both boys and girls [25]. Nevertheless, studies have demonstrated that approximately 50–65% of children and adults do not achieve the European Food Safety Association (EFSA) recommendation for total drinking fluids [26,27]. In China, about 32% of adults aged 18–60 years old and adolescents drank less water than that recommended by the Chinese Nutrition Society in 2012 [28]; furthermore, among young adults, about 80.1% of them did not drink an adequate amount of fluids [29], which increases the risk of dehydration. Studies have revealed that dehydration impairs physical performance, and even cognitive performance [30,31,32]. Therefore, the insufficient intake of water may have adverse effects on body composition. The effects of insufficient water intake on body composition are poorly understood and evidence is both scarce and scattered. Furthermore, only one study evaluating the water intake and body composition has been performed among young adults in free-living conditions in China [24]. However, there is a lack of information about the effects of water restriction or water supplementation on body composition among young adults globally, and especially in China, and therefore more studies need to be carried out.

The aims of this study were (1) to investigate the effects of water restriction on body composition; (2) to explore changes of water supplementation on body composition; (3) to examine the optimum volume of water that improves body composition after water restriction among young adults aged 18–23 years old in China.

## 2. Methods

### 2.1. Study Design

This was a randomized controlled trial that lasted for 3 days.

### 2.2. Sample Size Calculation

It was reported that the TBW of young males with euhydration and dehydration conditions were 48.8 and 45.4 L, and the standard deviations were 7.5 and 7.3 L, respectively [33]. Software PASS 11.0 (NCSS, LLC, Kaysville, UT) was used to calculate the sample size for the differences in body composition among participants. Statistical significance α was set at 0.05 (*p* < 0.05, 2-tailed), power (1-Beta) was 0.90. Additionally, a 10% drop-out rate was considered. A total of 64 participants were needed in this study.

### 2.3. Study Participants

The inclusion criteria were that participants were healthy and 18–23 years old. The exclusion criteria were that participants who habitually smoked, consumed alcohol (>20 g/day), had high caffeine consumption (>250 mg/day), or had chronic or other diseases were not included in this study [34]. A total of 76 young adults were recruited in this study, including 40 males and 36 females.

### 2.4. Ethics

The study protocol and instruments were reviewed and approved by the Peking University Institutional Review Committee. The ethical approval identification code is IRB00001052–16071. The study was conducted according to the guidelines of the Declaration of Helsinki. A signed informed consent form was obtained from each participant prior to the study.

### 2.5. Study Design and Procedure

A randomized controlled trial study was conducted, which lasted for three days.

On day 1, all participants were asked to fast overnight for 12 h, from 8:00 p.m. of day 1 to 8:00 a.m. of day 2, without having any food or drink. On day 2, at 8:00 a.m., a set of baseline tests, including anthropometric indices (height, weight, and body composition) and urine and blood samples (plasma osmolality, urine osmolality, and urine specific gravity), was done—measured by trained investigators. Then, all participants were instructed to restrict water intake for 24 h, but with three meals containing ≤75% of water provided on day 2. All foods were weighed before and after the participants ate to assess the amount of water in the food. On day 3, they were asked to arrive in the laboratory at 8:00 a.m. The same indices were measured as for the dehydration test. Then, participants were randomly separated into four groups: WR groups 1, 2, and 3 (water replenishment groups given 1000, 500, and 200 mL purified water, respectively) and a NR group (no water replenishment group with no water). At 8:30 a.m., participants in WR groups 1, 2, and 3 were asked to drink the corresponding volumes of purified water within 10 min, and the participants in the NR group did not drink water or any other fluids. The water was provided in three opaque containers. After water supplementation for 90 min (at 10:00 a.m.), weight, body composition, urine, and blood samples were tested as a rehydration test. The study procedure is shown in Figure 1.

### 2.6. Assessment of Water from Food

On day 2, participants were asked to have only the foods with water content <75% that were provided by the researchers. The foods were weighed before and after the participants ate. All foods consumed by the participants were weighed accurately by trained investigators using an electronic balance (YP20001; SPC; Shanghai, China).

### 2.7. Temperature and Humidity of the Environment

The temperature and humidity of the study locations indoors and outdoors were recorded every day at three time points, 10:00 a.m., 2:00 p.m., and 8:00 p.m., for 3 days (WRB-1-H2, Exasace, Zhengzhou, China).

### 2.8. Anthropometry

Height was measured twice to the nearest 0.1 cm using a height gauge with participants in light clothing to the nearest 0.1 kg by trained investigators with standard procedures (HDM-300; Huaju, Zhejiang, China). Weight was measured twice using weight meter by an experienced investigator. The measured values of height and weight were averaged. (BMI: weight (kg)/height squared (m)).

Body composition was measured using a bioelectrical impedance analyzer (Inbody 720; Inbody; Seoul, Korea) by trained investigators, with participants in light clothing and barefoot. Based on the theory of four component model of body composition, the instrument uses 8-point contact electrodes (two thumb electrodes, two palm electrodes, two sole electrodes and two heel electrodes) to measure 30 impedance values at 5 segments (left and right upper limbs, trunk, and lower limbs) at 6 different frequencies (1, 5, 50, 250, 500, and 1000 kHz). Inbody 720 uses different high-frequency and low-frequency conditions to measure the intracellular and extracellular water, so as to accurately analyze the total water content.

Participants were asked to place their feet on the foot electrode properly, and to hold the hand electrode with both hands. The angle between the trunk and upper limbs of the participants was 15 degrees. After inputting the index, age, height, and gender of the participants into the Inbody 720, the total body water (TBW), intracellular water (ICW), and extracellular water (ECW) were measured for about 2 min.

Bioelectrical impedance analyzers (BIAs) are widely used for evaluation of body compartments in many clinical and non-clinical settings [35]. BIAs are considered to be an easily, noninvasive, and quickly applied method and provide reliable outcomes in research and clinical trials. The BIA method can separately measure total body water (TBW), intracellular water (ICW), extracellular water (ECW), and other body compartments. Validation of BIA for the effective measurement of body compositions has been shown in the results of some studies [36,37,38].

### 2.9. Urine and Plasma Biomarkers

The urine and blood samples were collected and tested. The urine samples were stored at +4 °C before measurement. The blood samples were centrifuged and the supernatants were placed in a refrigerator for further determination. Urine osmolality and plasma osmolality were determined via the freezing point method using an osmotic pressure molar concentration meter (SMC 30C; Tianhe, Tianjin, China). USG (urine specific gravity) measurement was conducted using an automatic urinary sediment analyzer with the uric dry-chemistry method (H-800; Dirui, Changchun, China).

### 2.10. Statistics

All statistical analyses were performed using SPSS 21.0 software (IBM Corp., Armonk, NY, USA). Results are presented as mean ± standard deviation (SD). The hydration status results are shown as numbers and percentages. Differences between variables were assessed with one-way ANOVA and the Chi-squared test. The differences between baseline test and dehydration test were compared using Student’s paired *t*-test. Repeated measurement data analysis was used to explore the effects of water supplementation on body composition among the four groups. The Kruskal–Wallis test was used to compare the differences in the hydration status among the four groups for baseline, dehydration, and rehydration tests. If the one-way ANOVA or Kruskal–Wallis test was significant, a post-hoc analysis was performed to determine which groups differed from other groups. The Mann–Whitney U-test and Student’s *t*-test were used to compare the differences between males and females within the same group in the baseline, dehydration, and rehydration tests. A *p*-value of less than 0.05 was considered an indication of a statistically significant result.

## 3. Results

### 3.1. Characteristics of Participants

In total, 76 participants were recruited in the study and 76 completed the study, with a completion rate of 100%. As shown in Appendix A, there were no significant differences in their characteristics, including age, height, weight, and BMI among the four groups (*p* > 0.05). Furthermore, the temperature and humidity of the study places outdoors and indoors during the study was shown in Appendix A; the thirst, urine and plasma biomarkers of participants was shown in Appendix A; the fluids from food, 24 h urine volume, void number and urine osmolality on day 2 of participants was shown in Appendix A.

### 3.2. Water Supplementation Effects on Body Composition

As shown in Table 1, when comparing the baseline with the dehydration tests among all the participants, the ICW/TBW increased, and the ECW, ECW/TBW, ECW/ICW, and TBW all decreased (all *p* < 0.05). When comparing dehydration test with rehydration test in the four groups, significant interactions between time and volume were found in ICW and TBW (*F* = 3.002, *p* = 0.036; *F* = 2.907, *p* = 0.040), and the ECW tended to be significantly different (*F* = 2.592, *p* = 0.059) among the four groups. In the rehydration test, when comparing WR group 1 with the NR group, significant differences were found in ICW and TBW (*p* = 0.031; *p* = 0.047); significant differences were found in ICW between WR group 2 and the NR group (*p* = 0.031), and the TBW approached significance (*p* = 0.058); however, no significant differences were found in ICW and TBW between WR group 3 and the NR group (all *p* > 0.05). When comparing WR groups 1 and 2, the ICW did not differ significantly (*p* > 0.05). When comparing the dehydration test with the rehydration test within each group, in WR group 1, the increases of ICW and TBW did not differ significantly (*p* = 0.579; *p* = 0.427); for WR groups 2 and 3 and the NR group, significant decreases were found in ICW (*p* = 0.037; *p* = 0.004; *p* < 0.001), and significant reductions were also found in TBW (*p* = 0.047; *p* = 0.003; *p* < 0.001), as shown in Table 1.

As shown in Table 2, in males, when comparing baseline with dehydration tests, significant differences were found in ECW, ICW/ECW, ICW/TBW, and ECW/TBW (all *p* < 0.05), and the ICW, TBW, and TBW/BW did not differ significantly (all *p* > 0.05). After water replenishment, significant interaction between time and volume was only found in ICW (*F* = 3.061, *p* = 0.040). In the rehydration test, a significant difference was found in ICW when comparing WR group 1 with the NR group (*p* = 0.007), but no significant differences were found between WR groups 2 and 3, and NR group, respectively (*p* = 0.052; *p* = 0.115). When comparing dehydration with rehydration tests, a significant reduction of 0.2 kg was found in ICW in the NR group (*p* = 0.009), and there were decreases in WR groups 2 and 3, but no significant differences (*p* = 0.842; *p* = 0.212). In WR group 1, there was an increase in ICW, but this was not significant (*p* = 0.846).

As shown in Table 2, in females, when comparing baseline with dehydration tests, significant differences were found in ICW, ECW, TBW, ICW/ECW, ICW/TBW, and ECW/TBW (all *p* < 0.05), and no significant difference was found in TBW/BW (*p* > 0.05). After water replenishment, significant interactions between time and volume were found in ICW and TBW (*F* = 3.002, *p* = 0.036; *F* = 2.907, *p* = 0.040). In the rehydration test, no statistically significant differences were found in ICW and TBW when comparing WR groups 1, 2, and 3 with the NR group (all *p* > 0.05), respectively. The ICW (−0.3 kg, −0.3 kg and −0.4 kg) and TBW (−0.4 kg, −0.5 kg and −0.5 kg) contents during the rehydration test were much lower in WR groups 2 and 3 and the NR group than that tested during the dehydration test (all *p* < 0.05). There were even increases of 0.2 kg and 0.5 kg in ICW and TBW in WR group 1 respectively, but with no statistical significance (*p* = 0.613; *p* = 0.462).

Results in Table 2 demonstrated that significant differences of body composition were found between males and females. When comparing males with females totally, in baseline and dehydration tests, males had higher ICW, ECW, TBW, and TBW/BW, and lower ICW/TBW, ECW/TBW, and ICW/ECW than females (all *p* < 0.05). When comparing the differences between males and females in each group, in the baseline and dehydration tests, in WR groups 1 and 2, significant differences between males and females were found in ICW, ECW, ICW/TBW, ECW/TBW, ICW/ECW, TBW, and TBW/BW (all *p* < 0.05). Furthermore, in WR group 3 and the NR group, significant differences were found in ICW, ECW, ICW/TBW, ECW/TBW, ICW/ECW, and TBW (all *p* < 0.05), respectively. In the rehydration test, in WR groups 1 and 2, and NR, significant differences between males and females were found in ICW, ICW/TBW, ECW, ECW/TBW, ECW/ICW, TBW, and TBW/BW (all *p* < 0.05). In WR group 3, significant differences were found between males and females in ICW, ICW/TBW, ECW, ECW/TBW, ECW/ICW, TBW, and TBW/BW (all *p* < 0.05), and the TBW/BW tended to have significance (*p* = 0.062).

## 4. Discussion

To our knowledge, the current study is the first to investigate the effects of water restriction for 36 h and different amounts of water replenishment on body composition among young adults in China.

Regarding the effects of water restriction of 36 h on body composition, significant decreases in ECW, ECW/TBW, and ECW/ICW, and an increase in ICW/TBW, were found among the young adults, compared with water restriction for 12 h. Furthermore, in males, after water restriction for 36 h, except for an increase in ICW/TBW, the ECW, ECW/TBW, and ECW/ICW all decreased; however, in females, significant reductions in the ICW, ECW, ECW/TBW, ECW/ICW, and TBW, including an increase in ICW/TBW, were observed. We can conclude that water restriction can change the body composition and young females are more susceptible to the effects of water restriction than males. Consistent with the current findings, a study conducted in rats revealed that after acute absolute body-fluid deficits, an about 2% reduction in ICW was induced [39]. Regarding the studies implemented among humans, one study conducted among triathletes observed that the TBW, ECW, and ICW all decreased significantly after the ultra-endurance triathlon race [40]. Another study confirmed the decrease in ECW from 18.5 to 18.2 kg among young male wrestlers [41]. However, the results of some studies are inconsistent with the current study. A randomized controlled study conducted among healthy males showed that the TBW and ECW both increased, but the ICW did not change significantly after 6 days of heat acclimation (39.8 °C, relative humidity 59.2%) from the control baseline [42]. Furthermore, studies evaluating exercise and TBW among young adults revealed that the TBW increased by 2.4 or 6.1% by the end of the races [43,44]. No other related studies exploring water restriction and body composition have been reported. Due to the different methods that lead to the changes of the body composition, the aspects of the body composition that change may be different. In the current study, except for the ECW/TBW and ECW/ICW, the level of body water was higher in males than for females during the baseline test and dehydration test. These results are in line with those of other studies that found differences between males and females. In free-living conditions, the body composition, including ICW, ICW/TBW, ECW, ECW/TBW, TBW, and TBW/BW, was higher among young Chinese males than young Chinese females [21]. Moreover, studies showed that there are many factors affecting body composition, including gender, age, race, body size, physical activity, and region [6,45,46,47]. Differences have even been found in the same participants when evaluating body composition with supine and standing shifts [48]. Totally, the water restriction for 36 h impedes the indices of body composition, including ICW, ECW, and TBW, among young adults. In addition, more adverse effects are found in females than in males. In free-living conditions, females should pay more attention to the amount of water intake, in order take in a sufficient amount.

Our main result was the significant improvements in the ICW and TBW after water replenishment, in which participants had higher ICW and TBW when they received 1000 mL water than those with no water. Furthermore, significant decreases were found in ICW and TBW in participants who drank no water, and even in those they drank 500 or 200 mL, which meant that only a large amount of water could alleviate the further damage of 36 h water restriction on the ICW and TBW. For males, after water replenishment, only participants who drank 1000 mL had higher ICW than those drank no water. However, for females, those who consumed different amounts of 1000, 500, and 200 mL of water did not have higher ICW and TBW than those who drank nothing. For a better understanding of the differences between males and females in our study, they can be compared with the results of other studies. It was revealed that after water consumption of 500 mL, the TBW of females did not differ significantly from the baseline, but an increase in males was observed [49]. Moreover, these results in the current study were in line with other studies that found similar increases in ICW after water replenishment. A randomized controlled design study employed in rats revealed that reduction in ICW was apparently recovered after water and NaCl consumption [39]. Another study revealed that the ICW contribution to plasma volume restoration became more pronounced as acclimation progressed after water replenishment of 200 mL among young males [42]. This indicated that only a large amount of water (1000 mL) could improve body composition for males after water restriction for 36 h. The amount of water of 500 or 200 mL did not alleviate the decreases in ICW and TBW for males or females. Moreover, for females, a larger amount of water and a longer intervention time would be required to substantially change the adverse effects of water restriction on body composition.

In our study, after water replenishment, we observed that the TBW was higher in adults who consumed 1000 mL than in those who drank no water, while no differences were found among participants consuming 500 or 200 mL when compared to those who drank nothing. Furthermore, the TBW all decreased in those who drank 500 or 200 mL and in those who drank no water. Therefore, we concluded that only a large amount of water consumption can improve the TBW of adults after 36 h of water restriction. There are several other findings in the literature that are compatible with this interpretation. One study conducted among healthy young women with TWI below 2L/d demonstrated that a sustained increase in total drinking fluids of +0.5 L/day for 6 weeks might result in slow recovery of TBW from an initial <3% TWI deficit [50]. Additionally, one study confirmed the increases in TBW after four periods of acute water consumption of 500 mL compared with the baseline among males and females [49]. In another study, the TBW/BW increased significantly after water supplementation with two additional 550 mL bottles of water for 12 weeks [51]. Some studies even evaluated the associations between TWI and TBW among adults in free-living conditions. A study conducted among adults in Spain revealed that drinking water correlated with TBW in males (*r* = 0.196) and in females (*r* = 0.187) [52]; moreover, among young males and females in China, significant associations were also found between TWI and TBW [24]. In contrast to the findings mentioned above, some studies did not observe improvements in body composition including the TBW after water supplementation. A study evaluating the TBW response to ad libitum water replacement using two different water delivery systems during a 19 km route march showed that water intake of 538 and 533 mL/h resulted in no differences in TBW [53]. Furthermore, a randomized controlled design conducted among young healthy males revealed no changes in TBW after fluid consumption for two weeks [54]. Additionally, another study implemented among adults after completing a race within 30 h demonstrated that no significant changes of TBW and ECW were observed pre- or post-race [55]. Different methods and duration of water intervention and various TBW measurements may be attributed to the different results of these studies. In the future, more studies should be conducted in this issue.

Our research had some strengths. Firstly, in this study, a randomized controlled design was used to reduce the bias. Secondly, the osmolality of urine was measured to monitor the hydration status of participants during the study. However, the present study had certain limitations. Firstly, the impacts of water supplementation for long-term intervention on body composition were not explored. Additionally, biomarkers of the hydration status (such as copeptin) were not measured in this study.

## 5. Conclusions

Water restriction has adverse effects on body composition, and females are more susceptible to water restriction than males. Water replenishment improves the water content of body composition, alleviating the effects of water restriction on ICW and TBW. The amount of 1000 mL is the optimum volume to improve the body composition among young male adults, but not females, after water restriction of 36 h.

## Figures and Tables

**Figure 1 nutrients-13-00553-f001:**
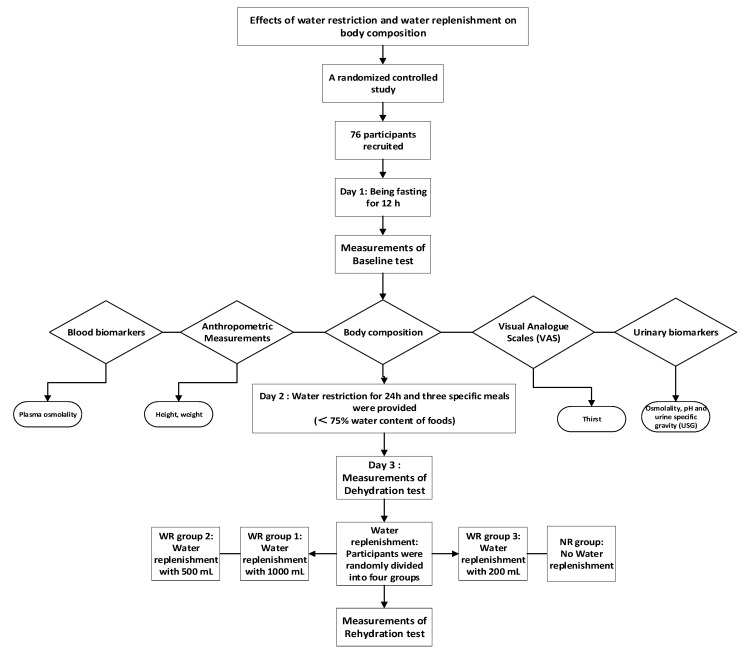
Study procedure.

**Table 1 nutrients-13-00553-t001:** Body composition of participants.

	Baseline Test					Dehydration Test					*t*	*p*	Rehydration Test				*F*	*p*
Total	WR Group 1 (*n* = 20)	WR Group 2 (*n* = 20)	WR Group 3 (*n* = 18)	NR Group (*n* = 18)	Total	WR Group 1 (*n* = 20)	WR Group 2 (*n* = 20)	WR Group 3 (*n* = 18)	NR Group (*n* = 18)	Total			WR Group 1 (*n* = 20)	WR Group 2 (*n* = 20)	WR Group 3 (*n* = 18)	NR Group (*n* = 18)		
ICW	21.4 ± 3.6	21.3 ± 4.8	20.1 ± 4.1	20.8 ± 3.7	20.9 ± 4.0	21.5 ± 3.8	21.1 ± 4.8 ^a^	20.1 ± 4.1 ^a^	20.7 ± 3.7 ^a^	20.9 ± 4.1	1.347	0.182	21.5 ± 3.7	21.0 ± 4.8	19.9 ± 4.2	20.4 ± 3.7	3.002	0.036
ICW/TBW (%)	62.2 ± 0.6	62.4 ± 0.6	62.3 ± 0.6	62.4 ± 0.6	62.3 ± 0.6 ^a^	62.5 ± 0.7 ^a^	62.6 ± 0.5	62.7 ± 0.7	62.8 ± 0.7 ^a^	62.7 ± 0.6	−9.937	<0.001	62.3 ± 0.8	62.5 ± 0.6	62.6 ± 0.7	62.6 ± 0.8	1.073	0.366
ECW	13.0 ± 2.0	12.8 ± 2.7	12.1 ± 2.2	12.5 ± 2.1	12.6 ± 2.3 ^a^	12.9 ± 2.1	12.6 ± 2.7	11.9 ± 2.2 ^a^	12.3 ± 2.1 ^a^	12.4 ± 2.3	6.946	<0.001	13.0 ± 2.0	12.5 ± 2.7	11.8 ± 2.3	12.2 ± 2.1	2.592	0.059
ECW/TBW (%)	37.8 ± 0.5	37.6 ± 0.5	37.8 ± 0.6	37.6 ± 0.7	37.7 ± 0.6 ^a^	37.5 ± 0.7 ^a^	37.4 ± 0.5	37.3 ± 0.7	37.2 ± 0.7 ^a^	37.4 ± 0.6	9.936	<0.001	37.7 ± 0.8	37.4 ± 0.6	37.4 ± 0.7	37.4 ± 0.8	1.076	0.365
ECW/ICW (%)	60.8 ± 1.5	60.2 ± 1.5	60.5 ± 1.5	60.2 ± 1.8	60.4 ± 1.6 ^a^	60.1 ± 1.7 ^a^	59.7 ± 1.3	59.5 ± 1.7	59.3 ± 1.8 ^a^	59.6 ± 1.6	9.926	<0.001	60.4 ± 2.1	60.0 ± 1.4	59.7 ± 1.9	59.8 ± 2.0	1.056	0.373
TBW	34.5 ± 5.6	34.0 ± 7.4	32.2 ± 6.3	33.3 ± 5.8	33.5 ± 6.3 ^a^	34.3 ± 5.9	33.7 ± 7.5 ^a^	32.0 ± 6.3 ^a^	33.0 ± 5.8 ^a^	33.3 ± 6.4	3.768	<0.001	34.5 ± 5.6	33.5 ± 7.5	31.7 ± 6.5	32.6 ± 5.8	2.907	0.040
TBW/BW (%)	54.3 ± 6.8	52.4 ± 5.7	55.3 ± 4.9	54.8 ± 4.1	54.9 ± 5.5	54.3 ± 7.1 ^a^	52.4 ± 5.7 ^a^	55.1 ± 4.9 ^a^	54.7 ± 3.7 ^a^	54.1 ± 5.5	0.942	0.349	53.6 ± 7.2	52.0 ± 5.7	54.6 ± 4.9	54.2 ± 3.8	0.594	0.621

Note: Values are shown as the mean ± standard deviation (SD). ^a^
*p* < 0.05 in the comparison between the baseline and dehydration tests. ^b^
*p* < 0.05 in the comparison between the dehydration and rehydration test. Compared with the dehydration test, during the rehydration test, in WR group 1, significant differences were found in ICW/TBW, ECW/TBW, ECW/ICW, and TBW/BW (*t* = 2.636, *p* = 0.016; *t* = −2.636, *p* = 0.016; *t* = −2.650, *p* = 0.016; *t* = 3.157, *p* = 0.005); in WR group 2, significant differences were found in ICW, TBW/BW, and TBW (*t* = 2.245, *p* = 0.037; *t* = 3.069, *p* = 0.006; *t* = 2.127, *p* = 0.047); in WR group 3, significant differences were found in ICW, ECW, TBW/BW, and TBW (*t* = 3.376, *p* = 0.004; *t* = 2.779, *p* = 0.013; *t* = 3.218, *p* = 0.005; *t* = 3.427, *p* = 0.003); in the NR group, significant differences were found in ICW, ICW/TBW, ECW, ECW/TBW, ECW/ICW, TBW, and TBW/BW (*t* = 5.732, *p* < 0.001; *t* = 3.921, *p* = 0.001; *t* = 2.496, *p* = 0.023; *t* = −3.921, *p* = 0.001; *t* = −3.922, *p* = 0.001; *t* = 5.052, *p* < 0.001; *t* = 3.801, *p* = 0.001). During baseline and dehydration tests, there were no significant differences found in ECW/ICW, TBW/BW, ICW/TBW, or ECW/TBW among the four groups (*F* = 0.695, *p* = 0.558; *F* = 0.963, *p* = 0.415; *F* = 0.703, *p* = 0.553; *F* = 0.703, *p* = 0.553; *F* = 0.833, *p* = 0.480; *F* = 0.915, *p* = 0.438; *F* = 0.843, *p* = 0.475; *F* = 0.843, *p* = 0.475). During the baseline test, no significant differences were found in ICW, ECW, or TBW among the four groups (*F* = 0.399, *p* = 0.754; *F* = 0.539, *p* = 0.657; *F* = 0.446, *p* = 0.721). During the dehydration test, no significant differences were found in ICW, ECW, or TBW among the four groups (*F* = 0.382, *p* = 0.766; *F* = 0.600, *p* = 0.617; *F* = 0.455, *p* = 0.715).

**Table 2 nutrients-13-00553-t002:** Body compositions of males and females.

	Baseline Test					Dehydration Test					*t*	*p*	Rehydration Test				*F*	*p*
Males	WR Group 1 (*n* = 10)	WR Group 2 (*n* = 10)	WR Group 3 (*n* = 10)	NR Group (*n* = 10)	Total	WR Group 1 (*n* = 10)	WR Group 2 (*n* = 10)	WR Group 3 (*n* = 10)	NR Group (*n* = 10)	Total			WR Group 1 (*n* = 10)	WR Group 2 (*n* = 10)	WR Group 3 (*n* = 10)	NR Group (*n* = 10)		
ICW	24.4 ± 1.5 ^c^	24.7 ± 4.2 ^c^	23.4 ± 2.7 ^c^	23.5 ± 2.3 ^c^	24.0 ± 2.8 ^d^	24.5 ± 1.7 ^c^	24.5 ± 4.4 ^c^	23.4 ± 2.6 ^c^	23.4 ± 2.4 ^bc^	24.0 ± 2.9 ^d^	0.268	0.790	24.5 ± 1.8 ^c^	24.5 ± 4.3 ^c^	23.3 ± 2.6 ^c^	23.2 ± 2.3 ^c^	3.061	0.040
ICW/TBW (%)	62.5 ± 0.4 ^c^	62.8 ± 0.4 ^c^	62.7 ± 0.3 ^c^	62.8 ± 0.5 ^c^	62.7 ± 0.4 ^ad^	62.8 ± 0.4 ^c^	62.9 ± 0.3 ^c^	63.1 ± 0.4 ^c^	63.2 ± 0.6 ^bc^	63.0 ± 0.5 ^d^	−7.291	<0.001	62.8 ± 0.4 ^c^	62.9 ± 0.3 ^c^	62.1 ± 0.5 ^c^	63.0 ± 0.6 ^c^	2.702	0.060
ECW	14.6 ± 0.9 ^c^	14.6 ± 2.5 ^c^	13.9 ± 1.5 ^c^	13.9 ± 1.6 ^c^	14.3 ± 1.7 ^ad^	14.5 ± 1.0 ^c^	14.4 ± 2.6 ^c^	13.7 ± 1.5 ^c^	13.7 ± 1.5 ^c^	14.1 ± 1.7 ^d^	3.979	<0.001	14.5 ± 1.0 ^c^	14.4 ± 2.5 ^c^	13.6 ± 1.5 ^c^	13.6 ± 1.5 ^c^	1.475	0.238
ECW/TBW (%)	37.5 ± 0.4 ^c^	37.2 ± 0.4 ^c^	37.2 ± 0.3 ^c^	37.2 ± 0.5 ^c^	37.3 ± 0.4 ^ad^	37.2 ± 0.4 ^c^	37.1 ± 0.3 ^c^	36.9 ± 0.4 ^c^	36.8 ± 0.6 ^bc^	37.0 ± 0.5 ^d^	7.291	<0.001	37.2 ± 0.4 ^c^	37.1 ± 0.3 ^c^	37.9 ± 0.5 ^c^	37.0 ± 0.6 ^c^	2.702	0.060
ECW/ICW (%)	60.1 ± 1.1 ^c^	59.2 ± 1.1 ^c^	59.3 ± 0.9 ^c^	59.2 ± 1.3 ^c^	59.5 ± 1.1 ^ad^	59.2 ± 1.0 ^c^	58.9 ± 0.9 ^c^	58.4 ± 1.1 ^c^	58.3 ± 1.4 ^bc^	58.7 ± 1.1 ^d^	7.338	<0.001	59.2 ± 1.1 ^c^	59.0 ± 0.7 ^c^	58.4 ± 1.2 ^c^	58.7 ± 1.6 ^c^	2.697	0.060
TBW	39.0 ± 2.4 ^c^	39.3 ± 6.7 ^c^	37.2 ± 4.1 ^c^	37.4 ± 3.8 ^c^	38.3 ± 4.4 ^d^	39.0 ± 2.6 ^c^	38.9 ± 7.0 ^c^	37.1 ± 4.1 ^c^	37.1 ± 3.8 ^bc^	38.1 ± 4.6 ^d^	1.772	0.084	39.0 ± 2.8 ^c^	38.9 ± 6.8 ^c^	36.9 ± 4.1 ^c^	36.8 ± 3.7 ^c^	2.500	0.075
TBW/BW (%)	58.5 ± 5.1 ^c^	55.5 ± 5.7 ^c^	57.2 ± 5.0	56.3 ± 4.7	56.9 ± 5.1 ^d^	58.7 ± 5.4 ^bc^	55.5 ± 5.9 ^c^	57.0 ± 4.8	56.1 ± 4.3 ^b^	56.8 ± 5.1 ^d^	0.419	0.677	58.2 ± 5.3 ^c^	55.3 ± 5.7 ^c^	56.7 ± 4.7	55.8 ± 4.3 ^c^	0.628	0.602
Females	WR group 1 (*n* = 10)	WR group 2 (*n* = 10)	WR group 3 (*n* = 8)	NR group (*n* = 8)	Total	WR group 1 (*n* = 10)	WR group 2 (*n* = 10)	WR group 3 (*n* = 8)	NR group (*n* = 8)	Total	*t*	*p*	WR group 1 (*n* = 10)	WR group 2 (*n* = 10)	WR group 3 (*n* = 8)	NR group (*n* = 8)	*F*	*p*
ICW	17.9 ± 1.4	17.8 ± 2.0	16.8 ± 2.1	17.4 ± 1.7	17.5 ± 1.8 ^a^	17.7 ± 1.4	17.8 ± 1.9 ^b^	16.7 ± 2.0 ^b^	17.4 ± 1.7 ^b^	17.4 ± 1.8	2.622	0.013	17.9 ± 0.8	17.5 ± 1.9	16.4 ± 2.1	17.0 ± 1.7	3.002	0.036
ICW/TBW (%)	61.8 ± 0.6	62.1 ± 0.5	61.9 ± 0.4	61.9 ± 0.6	61.9 ± 0.5 ^a^	62.1 ± 0.7 ^b^	62.4 ± 0.5 ^b^	62.3 ± 0.6 ^b^	62.3 ± 0.5 ^b^	62.3 ± 0.6	−6.777	<0.001	61.8 ± 0.6	62.2 ± 0.6	62.1 ± 0.5	62.0 ± 0.6	1.073	0.366
ECW	11.1 ± 1.0	10.9 ± 1.2	10.4 ± 1.2	10.7 ± 1.1	10.8 ± 1.1 ^a^	10.8 ± 1.0	10.7 ± 1.1 ^b^	10.1 ± 1.1	10.5 ± 1.2	10.5 ± 1.1	6.744	<0.001	11.1 ± 0.7	10.6 ± 1.1	10.0 ± 1.2	10.4 ± 1.1	2.592	0.059
ECW/TBW (%)	38.2 ± 0.6	38.0 ± 0.5	38.1 ± 0.4	38.1 ± 0.6	38.1 ± 0.5 ^a^	37.9 ± 0.7 ^b^	37.7 ± 0.5 ^b^	37.7 ± 0.5 ^b^	37.7 ± 0.5 ^b^	37.7 ± 0.6	6.776	<0.001	38.3 ± 0.7	37.8 ± 0.6	37.9 ± 0.5	38.0 ± 0.6	1.076	0.365
ECW/ICW (%)	61.8 ± 1.5	61.2 ± 1.3	61.6 ± 1.1	61.5 ± 1.5	61.5 ± 1.3 ^a^	61.1 ± 1.8 ^b^	60.4 ± 1.4 ^b^	60.5 ± 1.5 ^b^	60.5 ± 1.3 ^b^	60.6 ± 1.5	6.758	<0.001	62.0 ± 1.9	60.9 ± 1.5	61.1 ± 1.4	61.2 ± 1.4	1.056	0.373
TBW	28.9 ± 2.4	28.7 ± 3.2	27.2 ± 3.3	28.1 ± 2.8	28.3 ± 2.9 ^a^	28.5 ± 2.3	28.5 ± 3.0 ^b^	26.9 ± 3.1 ^b^	27.9 ± 2.9 ^b^	27.9 ± 2.8	4.864	<0.001	29.0 ± 1.4	28.1 ± 2.9	26.4 ± 3.2	27.4 ± 2.8	2.907	0.040
TBW/BW (%)	49.2 ± 5.1	49.4 ± 3.8	53.4 ± 4.2	53.0 ± 2.2	51.2 ± 4.3	49.0 ± 5.1	49.3 ± 3.5 ^b^	53.3 ± 4.5 ^b^	53.0 ± 1.9 ^b^	51.1 ± 4.3	1.148	0.259	48.0 ± 4.9	48.7 ± 3.6	52.5 ± 4.3	52.2 ± 2.0	0.594	0.621

Note: Values are shown as mean ± standard deviation (SD). ^a^
*p* < 0.05 in the comparison between the baseline and dehydration tests. ^b^
*p* < 0.05 in the comparison between the dehydration and rehydration test. ^c^
*p* < 0.05 in the comparison between males and females within group. ^d^
*p*<0.05 in the comparison between males and females totally in the baseline and dehydration tests.

## Data Availability

The data presented in this study are available on request from the corresponding author. The data are not publicly available due to privacy.

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
