# Peer review of "Effects of Water Restriction and Water Replenishment on the Content of Body Water with Bioelectrical Impedance among Young Adults in Baoding, China: A Randomized Controlled Trial (RCT)"

_nutrients, 2021, doi:10.3390/nu13020553_

Round 1
Reviewer 1 Report
Abstract: "vulnerary"- unsure if this is the right word to use in context.
Introduction: Authors did not define the term "young adult."
page 3 -
- "Studies reveled..." should read "Studies revealed..."
- "... was both scare and scattered." should read "... was both scarce and scattered"
Methods:
- 2.1 - study duration too short; rationale for duration not given
- 2.5 -
- "On day 3, ... to arrive the lab..." should read ""On day 3, ... to arrive in the lab..."
- "... indexes" = indices
- 2.8 - "The values of height were averaged the values..." should read "The measured values of height were averaged."
Results:
The font size (5.5) for supplementary tables' content is too small.
3.1 - "... with a complete rate ..." should read "... with a completion rate ..."
3.3
- Please define "void number of urine"
- "... as shown in Supplementary Table 3."
3.5 -
- "... as shown in Table 1, ..."
- " When comparing baseline test ...
- page 8: "As shown in Table 2, ..."
- page 8: consider replacing "when compared ..." with "when comparing"
Discussion:
- 3rd paragraph:
- "Regarding the body ..."
- "Studies showed that ... many factors affecting ..."
- 4th paragraph:
- "... their counterparts who drank nothing, ... those who drank no water, ..."
- "In a study ... it was revealed that after water consumption ..."
- last but one line - "8 males revealed that ..."
- 5th paragraph:
- "After water replenishment, ... that consumed ... than those who drank ..."
- "A study conducted among adults in Spain revealed that drinking water ..."
- "One study ealuating TBW ... revealed that ..."
- Last but one sentence: "... revealed no changes of TBW after fluid consumption ..."
- Strengths and limitations paragraph: "However, the pesent study had certain limitations."
- Conclusion: authors use "vulnerary" again. Did they mean "vulnerable"?
Author Response
Comments and Suggestions for Authors
Abstract: "vulnerary"- unsure if this is the right word to use in context.
Response: Thank you for your comments.
We have revised the “vulnerary” into “more susceptible to” in the Abstract (Line 52, Page 2).
Introduction: Authors did not define the term "young adult."
Response: Thank you for your comments.
We have made the revision accordingly in the Introduction Section (Line 146, Page 3). The “young adults” was defined as young adults aged 18-23 years old.
page 3 -
- "Studies reveled..." should read "Studies revealed..."
- "... was both scare and scattered." should read "... was both scarceand scattered"
Response: Thank you for your comments.
We have made the revision accordingly (Line 130,135, Page 3).
Methods:
- 1 - study duration too short; rationale for duration not given
Response: Thanks for your comments.
In order to investigate the water restriction and water supplementation on body composition, this study was designed.
In this study, the duration of the water restriction was 36h. After searching the related references about water restriction, we found that the duration was between 24 h and 37 h. Furthermore, the study that investigating the effects of dehydration induced by water restriction of 36 h on cognitive performances and mood was conducted among young male adults in China by our team in 2016 and we found that it was the most appropriate duration of water restriction to meet the research purposes and for participants to complete the study. Therefore, we chose the 36 h and the study was conducted according to the guidelines of the Declaration of Helsinki.
Therefore, a randomized controlled trial study was conducted, which lasted for three days. On day 1, all participants were asked to fast overnight for 12h, from 8:00 PM of day 1 to 8:00 AM of day 2, without having any food or drink. On day 2, at 8:00 AM, baseline test including anthropometric indices (height, weight, and body composition), urine and blood samples (plasma osmolality, urine osmolality, and urine specific gravity) were obtained and measured by trained investigators. Then, all participants were instructed to restrict water intake for 24 h, but with three meals containing ≤75% of water provided on day 2. All foods were weighed before and after the participants ate to assess the amount of water in the food. On day 3, they were asked to arrive in the lab at 8:00 AM. The same indices were measured as for the dehydration test. Then, participants were randomly separated into four groups: WR groups 1, 2, and 3 (Water replenishment groups given 1000, 500, and 200 ml purified water, respectively) and an NR group (No water replenishment group with no water). At 8:30 AM, participants in WR groups 1, 2, and 3 were asked to drink the corresponding volumes of purified water within 10 minutes, and the participants in the NR group did not drink water or any other fluids. The water was provided in three opaque containers. After water supplementation for 90 minutes (at 10:00 AM), weight, body composition, urine, and blood samples were tested as rehydration test.
Followed are the References:
[1] Szinnai G, Schachinger H, Arnaud M J, et al. Effect of water deprivation on cognitive-motor performance in healthy men and women[J]. Am J Physiol Regul Integr Comp Physiol. 2005, 289(1): R275-R280.
[2] Pross N, Demazieres A, Girard N, et al. Influence of progressive fluid restriction on mood and physiological markers of dehydration in women[J]. Br J Nutr. 2013, 109(2): 313-321.
[3] Shirreffs S M, Merson S J, Fraser S M, et al. The effects of fluid restriction on hydration status and subjective feelings in man[J]. Br J Nutr. 2004, 91(6): 951-958.
[4] Zhang N, Du S M, Zhang J F, et al. Effects of Dehydration and Rehydration on Cognitive Performance and Mood among Male College Students in Cangzhou, China: A Self-Controlled Trial[J]. International Journal of Environmental Research and Public Health, 2019, 16(11):1891.
- 5 -
- "On day 3, ... to arrive the lab..." should read ""On day 3, ... to arrive inthe lab..."
- "... indexes" = indices
- 8 - "The values of height were averaged the values..." should read "The measured values of height were averaged."
Response: We have made the revision accordingly in the Methods Section (Lines 180, 185-186, 215, Pages 4-5).
Results:
The font size (5.5) for supplementary tables' content is too small.
Response: We have made the revision accordingly, and the front size of the supplementary tables was changed between 5.5 to 8.0.
3.1 - "... with a complete rate ..." should read "... with a completion rate ..."
Response: Thank you for your comments.
We have made the revision accordingly in the Results Section (Line 276, Page 7).
3.3
- Please define "void number of urine"
- "... as shown in Supplementary Table 3."
Response: Thank you for your comments.
We have made the revision accordingly in the Results Section.
The “void number of urine” was defined as the void number of 24 h urine. The void number was starting from the second voiding of the first day and ending with the first voiding the morning of the second day (Lines 288-290, Page 7).
We have revised the “as showed in Supplementary Table 3” into “as shown in Supplementary Table 3” (Line 294, Page 7).
3.5 -
- "... as shown in Table 1, ..."
- " When comparingbaseline test ...
- page 8: "As shownin Table 2, ..."
- page 8: consider replacing "when compared ..." with "when comparing"
Response: Thank you for your comments.
We have made the revision accordingly, revising the “showed” into “shown” and revising the “when compared” into “when comparing” (Lines 311, 314, 341, Pages 7, 10).
Discussion:
- 3rd paragraph:
- "Regarding the body ..."
- "Studies showed that ... many factors affecting ..."
Response: Thank you for your comments.
We have made the revision accordingly in the Discussion Section (Lines 407, 438, Page 12-13).
- 4th paragraph:
- "... their counterparts who drank nothing, ... those who drank no water, ..."
- "In a study ... it was revealed that after water consumption ..."
- last but one line - "8 males revealed that ..."
Response: Thank you for your comments. We have made the revision accordingly in the Discussion Section.
We have revised the “Furthermore, the ICW of participants consuming 1000 mL and 500 mL were higher than their counterparts drank nothing, but participants with 200 mL water did not differ with those drank no water” into “ Furthermore, the ICW of participants consuming 1000 mL and 500 mL was higher than for their counterparts who drank nothing, but participants with 200 mL water did not differ from those who drank no water” (Lines 451-455, Page 13).
We have revised the “it revealed” into “it was revealed” (Line 462, Page 13).
We have revised the “8 males reveled that” into “8 males revealed that” (Lines 468-469, Page 13).
- 5th paragraph:
- "After water replenishment, ... that consumed ... than those who drank ..."
- "A study conducted among adults in Spain revealed that drinking water ..."
- "One study ealuating TBW ... revealed that ..."
- Last but one sentence: "... revealed no changes of TBW after fluid consumption ..."
- Strengths and limitations paragraph: "However, the pesent study had certain limitations."
Response: Thank you for your comments. We have made the revision accordingly (Lines 428-429, 477-478, 494, 505, 516-517, Pages 12-14).
- Conclusion: authors use "vulnerary" again. Did they mean "vulnerable"?
Response: Thank you for your comments.
In this study, after water restriction of 36 h, more changes of the indices among females were found than among males. Furthermore, after water supplementation, the improvements were not found among females, but the indices of the body composition among males were improved. Therefore, we concluded that the females were more susceptible to the adverse effects of water restriction than males.
We have revised the “vulnerary” into “more susceptible to” (Line 505, Page 14).

Reviewer 2 Report
First of all, the research topic is interesting and it is very well discussed in the paper. The paper add a great scientific approach to this subject related with the effects of water restriction and water replenishment on body composition. I considered the strengths of this paper the introduction that explains very well the importance of this research and the link between the water compartments and health is brilliant. Also, the discussion is clear and easy for the readers.
However, I'd like to add some concerns:
Minor concerns:
In the introduction section I felt the lack of some references. For example:
1) Introduction L 67: Please insert reference and please describe whether it is referring to the healthy adults.
2) Introduction L 92: Please insert reference in this sentence: "A series of literatures showed that the body composition were associated with health."
3) Introduction L 107: Please insert reference in this sentence: "Moreover, observing the changes of body composition was of vital importance of human health".
------
In the results section:
4) "3.3. Water from Food, 24 h Urine Volume, Void Number, Urine Osmolality USG and electrolytes concentrations on Day 2" - L:248 please mention whether the urine volume is 24h urine volume.
5) In the table contents, it would be easier to read if the authors could fit the numbers in the same line. Although I understand that due the quantity of information it could be hard.
------
Major concerns:
Throughout the paper, the authors refer several times the TBW, ICW and ECW and its ratios (ECW/TBW and ECW/ICW). Given the importance of these variables for the quality of this study, the methods section should have detailed information about the assessment of these variables. Particularly, since bioimpedance is not a reference method, more information about the bioimpedance system should be given.
Author Response
Comments and Suggestions for Authors
First of all, the research topic is interesting and it is very well discussed in the paper. The paper add a great scientific approach to this subject related with the effects of water restriction and water replenishment on body composition. I considered the strengths of this paper the introduction that explains very well the importance of this research and the link between the water compartments and health is brilliant. Also, the discussion is clear and easy for the readers.
However, I'd like to add some concerns:
Minor concerns:
In the introduction section I felt the lack of some references. For example:
- Introduction L 67: Please insert reference and please describe whether it is referring to the healthy adults.
Response: Thank you for your comments.
The reference had been added into the Introduction Section (Lines 67-68, Page 2).
Followed is the reference:
[1] Chinese Nutrition Society. Chinese dietary reference intakes 2013. Beijing: Science press; 2014. 48–51.
- Introduction L 92: Please insert reference in this sentence: "A series of literatures showed that the body composition were associated with health."
Response: Thank you for your comments.
We have revised accordingly, by adding the related references into the Introduction Section (Lines 94-95, Page 2).
Followed are the references:
[1] Łatka M, Wójtowicz K, Drozdz T , et al. Relationship between water compartments, body composition assessed by bioelectrical imped-ance analysis and blood pressure in school children.[J]. Przeglad Le-karski. 2016, 73, 1.
[2] Kozio-Kozakowska A. Body Composition and a School Day Hydration State among Polish Chil-dren—A Cross-Sectional Study. Int. J. Env. Res. Pub. He. 2020, 17, 7181.
[3] Verlaan, S.A.; Aspray, T.J.; Bauer, J.M.; Cederholm, T.; Hemsworth, J.; Hill, T.R.; McPhee, J.S.; Piasecki, M.; Seal, C.; Sieber, C.C.; Borg, S.T.; Wijers, S.L.; Brandt, K. Nutritional status, body composition, and quality of life in community-dwelling sarcopenic and non-sarcopenic older adults: A case-control study. Clin. Nutr. 2017, 36, 267-274.
- Introduction L 107: Please insert reference in this sentence: "Moreover, observing the changes of body composition was of vital importance of human health".
Response: Thank you for your comments.
We have made the revision accordingly in the Introduction Section, and added the references in the manuscript (Lines 110-112, Page 3).
Followed is the reference:
[1] Mikkola. T.M.; Kautiainen. H.; Bonsdorff. M.B.; Salonen. M.K.; Wasenius. N.; Kajantie. E.; Eriksson. J.G. Body composition and changes in health‑related quality of life in older age: a 10‑year follow‑up of the Helsinki Birth Cohort Study[J]. Qual. Life. Res. 2020, 29, 2039-2050.
------
In the results section:
- "3.3. Water from Food, 24 h Urine Volume, Void Number, Urine Osmolality USG and electrolytes concentrations on Day 2" - L:248 please mention whether the urine volume is 24h urine volume.
Response: Thank you for your comments.
We have revised the “the urine volume” into “the 24 h urine volume” (Line 291, Page 7).
- In the table contents, it would be easier to read if the authors could fit the numbers in the same line. Although I understand that due the quantity of information it could be hard.
Response: Thank you for your comments.
We have revised the table contents (including Table 1 and 2), in order to make the numbers in the same line.
------
Major concerns:
Throughout the paper, the authors refer several times the TBW, ICW and ECW and its ratios (ECW/TBW and ECW/ICW). Given the importance of these variables for the quality of this study, the methods section should have detailed information about the assessment of these variables. Particularly, since bioimpedance is not a reference method, more information about the bioimpedance system should be given.
Response: Thank you for your comments. We have made the revision accordingly in the Methods Section (Lines 217-243, Pages 5-6).
Body composition was measured using a bioelectrical impedance analyzer (Inbody 720; Inbody; Seoul, Korea) by trained investigators, with participants in light clothing and barefoot. Based on the theory of four component model of body composition, the instrument uses 8-point contact electrodes (two thumb electrodes, two palm electrodes, two sole electrodes and two heel electrodes) to measure 30 impedance values at 5 segments (left and right upper limbs, trunk and lower limbs) at 6 different frequencies (1kHz, 5KHz, 50KHz, 250kHz, 500KHz and 1000kHz). Inbody 720 uses different high-frequency and low-frequency conditions to measure the intracellular and extracellular water, so as to accurately analyze the total water content.
Participants were asked to place their feet on the foot electrode properly, and to hold the hand electrode with both hands. The angle between the trunk and upper limbs of the participants were at 15 degrees. After inputting the index, age, height, and gender of the participants into the Inbody 720, the total body water (TBW), intracellular water (ICW), and extracellular water (ECW) were measured for about 2 minutes.
Bioelectrical impedance analyzers (BIAs) are widely used for evaluation of body compartments in many clinical and non-clinical settings [32]. BIAs are considered to be an easily, noninvasively, and quickly applied method and provide reliable outcomes in research and clinical trials. The BIA method can separately measure total body water (TBW), intracellular water (ICW), extracellular water (ECW), and other body compartments. Validation of BIA on effective measurement of body compositions has been shown in the results of some studies.
Followed are the references:
[1] Khalil, S.F.; Mohktar, M.S.; Ibrahim, F. The theory and fundamentals of bioimpedance analysis in clini-cal status monitoring and diagnosis of diseases. Sensors (Basel). 2014, 14, 10895–10928.
[2] Wu, C.S.; Chen, Y.Y.; Chuang, C.L.; Chiang, L.M.; Dwyer, G.B.; Hsu, Y.L.; Huang, A.C.; Lai, C.L.; Hsieh, K.C. Predicting body composition using foot-to-foot bioelectrical impedance analysis in healthy Asian individuals. Nutr. J. 2015, 19, 52.
[3] Andreoli A, Garaci F, Cafarelli FP, et al. Body composition in clinical practice. Eur. J. Radiol. 2016, 85, 1461–1468.
[4] Ling, C.H.Y.; Craen, A.J.M.D.; Slagboom, P.E.; Gunn, D.A.; Stokkel, M.P.M.; Westendorp, R.G.J.; Maier, A.B. Accuracy of direct segmental multi-frequency bioimpedance analysis in the assessment of total body and segmental body composition in middle-aged adult population. Clin. Nutr. 2011, 30, 610-615.

Reviewer 3 Report
This paper addresses an important question, the possible effects exerted by the intake/restriction of water on some body composition parameters.
Having said that, I now wish to turn to what I perceive the major concerns of the paper to be:
- Supplementary material should provide additional useful information to complement the main text. Instead, in this case, data from supplementary tables is cited in the Results section.
- Results of statistical tests (p values, etc.) are crowded into notes below the tables and are difficult to read. The text from lines 349 to 395, for example, seems to constitute a single note to Table 2. This is not an acceptable way of presentation.
-In our opinion, there is also some aspect that needs to be clarified in statistical methods. You assert " Results were presented as median and quartile ranges " (lines 22-223), but the article and related tables show means and standard deviations, and I have not seen interquartile ranges. In addition, it is also unclear where chi-squared was applied, nor to which variables the Mann-Whitney U test and Kruskal-Wallis test were applied.
- In the discussion of the literature, the results from the various studies are reported analytically without attempting in any way to summarize them.
- The English of the manuscript needs to be improved.
Minor concerns:
-Specify the number of participants in the study in section 2.3.Study Participants.
-Line 200: you mention 7 days of environmental temperature and humidity assessment, while in the text below and in table S2 you refer to 3 days.
- Tables 1 and 2 are to be rearranged both in terms of the variables on the columns (all in bold or not) and of some values with decimals not on the same row.
Author Response
Comments and Suggestions for Authors
This paper addresses an important question, the possible effects exerted by the intake/restriction of water on some body composition parameters.
Having said that, I now wish to turn to what I perceive the major concerns of the paper to be:
- Supplementary material should provide additional useful information to complement the main text. Instead, in this case, data from supplementary tables is cited in the Results section.
Response: Thank you for your comments. We have made revision accordingly.
In this study, the objectives of this study was to investigate the effects of water restriction and water replenishment on the body composition, including the total body water (TBW), the ICW and ECW. Therefore, we recruited 76 young adults totally.
Studies showed that the temperature and humidity have effects on the body composition, therefore, we recorded the temperature and humidity for three times during the three study days. Furthermore, the thirst, the urinary and plasma biomarkers were used to make sure the hydration statuses of the participants, and to improve the compliance of the participants. In our study, after water restriction of 36h, the osmolality of urine and plasma all increased, and the feelings of thirst were higher, when comparing with baseline test. Furthermore, after different amounts of water replenishment, the osmolality of urine and plasma were all decreased in different degree and the feelings of thirst were also improved. While, the effects of different amounts of water replenishment on body composition did not have the same improvements.
Therefore, this indicated that even after water replenishment, the urinary and plasma osmolality decreased and the hydration statuses were improved, but the adverse effects of the water restriction on body composition did not improve, which may needs more time to recover to the original levels. We should pay more attention to the body composition and have adequate water intake in the free-living conditions.
Then, we added the information about the temperature and humidity of the study days, the characteristics of the young adults, and the thirst, the urinary and plasma biomarkers as Supplementary materials. Moreover, we mentioned the information in the Results Section with a brief description, and we wanted to provide more information of the study for other researchers and made it easier to follow.
But, we did not discuss much more about Supplementary materials in the Discussion Section.
- Results of statistical tests (p values, etc.) are crowded into notes below the tables and are difficult to read. The text from lines 349 to 395, for example, seems to constitute a single note to Table 2. This is not an acceptable way of presentation.
Response: Thank you for your comments. We have revised accordingly.
The table had been re-edited and the explanations such as the statistical values under the Table 2 were removed. We used the letters to express the statistically significant differences between different groups or within groups in Table 2.
-In our opinion, there is also some aspect that needs to be clarified in statistical methods. You assert " Results were presented as median and quartile ranges " (lines 22-223), but the article and related tables show means and standard deviations, and I have not seen interquartile ranges. In addition, it is also unclear where chi-squared was applied, nor to which variables the Mann-Whitney U test and Kruskal-Wallis test were applied.
Response: Thank you for your comments. We have made the revision accordingly in the Methods Section (Lines 256-270, Page 6).
Results were presented as the mean ± standard deviation (SD). The hydration status was shown as numbers and percentages. Differences between variables were assessed with One-way ANOVA and the chi-squared test. The differences between baseline test and dehydration test were compared using Student’s paired t-test. Repeated measurement data analysis was used to explore the effects of water supple-mentation on body composition among the four groups. The Kruskal–Wallis test was used to compare the differences in the hydration statuses among the four groups in baseline, dehydration, and rehydration tests. If the One-way ANOVA or Kruskal–Wallis test was significant, a post-hoc analysis was performed to determine which groups differed from each other group. The Mann–Whitney U-test and Student’s t-test were used to compare the differences between males and females within the same group in the baseline, dehydration, and rehydration tests.
- In the discussion of the literature, the results from the various studies are reported analytically without attempting in any way to summarize them.
Response: Thank you for your comments.
We have summarized the results of various studies in the Discussion Section (Lines 403-406, 445-449, 471-476, 509-512, Pages 12-14).
- The English of the manuscript needs to be improved.
Response: Thank you for your comments.
The manuscript has been carefully revised by the MDPI (English-26057), in order to make sure that there were no errors in the English language and grammars throughout the manuscript.
Minor concerns:
-Specify the number of participants in the study in section 2.3.Study Participants.
Response: Thank you for your comments.
We have made the revision accordingly (Lines 164-165, Page 4).
A total of 76 young adults were recruited in this study, including 40 males and 36 females.
-Line 200: you mention 7 days of environmental temperature and humidity assessment, while in the text below and in table S2 you refer to 3 days.
Response: Thank you for your comments.
We have revised mistakes in the Methods Section “2.7 Temperature and humidity of the environment” (Line 207-210, Page 5).
The study duration was 3 days; therefore, we measured the temperature and humidity of the three study days.
The temperature and humidity of the study locations indoors and outdoors were recorded every day at three time points: 10:00 AM, 2:00 PM, and 8:00 PM, for 3 days (WRB-1-H2, Exasace, Zhengzhou, China).
- Tables 1 and 2 are to be rearranged both in terms of the variables on the columns (all in bold or not) and of some values with decimals not on the same row.
Response: Thank you for your comments.
The variables on the columns of Table 1 and Table 2 were rearranged and all the values with decimals were in the same row.

Round 2
Reviewer 2 Report
Dear authors,
Once again, congratulations for your manuscript. This scientific work adds a great scientific approach to this topic.
I'am pleased to verify that all changes I have requested have been successfully completed.
Best regards
Author Response
Point-by-point response to comments
Reviewer 2
Open Review
(x) I would not like to sign my review report
( ) I would like to sign my review report
English language and style
( ) Extensive editing of English language and style required
( ) Moderate English changes required
(x) English language and style are fine/minor spell check required
( ) I don't feel qualified to judge about the English language and style
|
|
Yes |
Can be improved |
Must be improved |
Not applicable |
|
|
Does the introduction provide sufficient background and include all relevant references? |
( ) |
(x) |
( ) |
( ) |
|
|
|
Is the research design appropriate? |
(x) |
( ) |
( ) |
( ) |
|
|
Are the methods adequately described? |
(x) |
( ) |
( ) |
( ) |
|
|
Are the results clearly presented? |
(x) |
( ) |
( ) |
( ) |
|
|
Are the conclusions supported by the results? |
(x) |
( ) |
( ) |
( ) |
Comments and Suggestions for Authors
Dear authors,
Once again, congratulations for your manuscript. This scientific work adds a great scientific approach to this topic.
I'am pleased to verify that all changes I have requested have been successfully completed.
Best regards
Response: Thank you for your comments.
We have revised the Introduction Section of the manuscript, by adding the related references.

Reviewer 3 Report
Although I have observed an improvement in the manuscript, especially a more correct use of the English language, it still need more revision. The tables have been corrected and some comments have been acknowledged. However, the text is still extremely awkward both in the Results and Discussion sections. In particular, in spite of my previous remarks concerning the Discussion, several studies are still reported in detail. The authors have only partly made the required changes and, in any case, they perhaps did not understand that it was not necessary to delete the studies reported, but simply to summarise the results of several studies (without indicating, for example, the number of subjects) with reference to the main common or contrasting results obtained. Moreover, some of the results are still based on supplementary data, despite my observations. If these data are really important for the study, the tables should be reported in the main text, in my opinion. Finally, it should be noted that the text is still partly in the form of a note. For example, the BMI and its formula are given in square brackets without any connecting sentence (line 233). In short, there is still work to be done in order to achieve a fluent and publishable scientific text.
Author Response
Point-by-point response to comments
Reviewer 3
Open Review
(x) I would not like to sign my review report
( ) I would like to sign my review report
English language and style
( ) Extensive editing of English language and style required
( ) Moderate English changes required
(x) English language and style are fine/minor spell check required
( ) I don't feel qualified to judge about the English language and style
|
Yes |
Can be improved |
Must be improved |
Not applicable |
|
|
Does the introduction provide sufficient background and include all relevant references? |
( ) |
(x) |
( ) |
( ) |
|
Is the research design appropriate? |
( ) |
( ) |
(x) |
( ) |
|
Are the methods adequately described? |
( ) |
(x) |
( ) |
( ) |
|
Are the results clearly presented? |
( ) |
( ) |
(x) |
( ) |
|
Are the conclusions supported by the results? |
( ) |
(x) |
( ) |
( ) |
Comments and Suggestions for Authors
Although I have observed an improvement in the manuscript, especially a more correct use of the English language, it still need more revision. The tables have been corrected and some comments have been acknowledged. However, the text is still extremely awkward both in the Results and Discussion sections. In particular, in spite of my previous remarks concerning the Discussion, several studies are still reported in detail. The authors have only partly made the required changes and, in any case, they perhaps did not understand that it was not necessary to delete the studies reported, but simply to summarise the results of several studies (without indicating, for example, the number of subjects) with reference to the main common or contrasting results obtained. Moreover, some of the results are still based on supplementary data, despite my observations. If these data are really important for the study, the tables should be reported in the main text, in my opinion. Finally, it should be noted that the text is still partly in the form of a note. For example, the BMI and its formula are given in square brackets without any connecting sentence (line 233). In short, there is still work to be done in order to achieve a fluent and publishable scientific text.
Response: Thank you for your comments.
We have discussed the Comments and Suggestions of yours one by one and made revision accordingly in the manuscript.
The manuscript has been carefully revised several times by our team and an English native speaker, in order to make sure that there were no errors in the English language and grammars throughout the manuscript.
We revised the Introduction Section, by adding related references; in the Methods Section, we made revision about the description of the study design; in the Results Section, we rearranged the description of Table 1 and Table 2, and made it clear for readers to understand the content in the table; in the Discussion Section, we summarized the related researches.
The objectives of the study were to investigate the effects of the water restriction and water replenishment on body composition among young adults (both in males and females), and to explore the optimal amount of water that improve the changes of body composition after water restriction of 36 h.
Regarding the Results and Discussion Sections, the main effects of the water restriction and water replenishment on body composition on all the participants (Table 1), on males and females (Table 2) were presented in order.
The effects of the water restriction and water replenishment on body composition on all the participants were shown in Table 1; the effects of water restriction and water replenishment on body composition on males and females were shown in Table 2.
Firstly, we compared the differences of body composition between baseline test and dehydration test, and we found that the water restriction impeded some aspects of the body composition. Then, the results of the comparison between dehydration test and rehydration test were presented as follow, which including the significant interactions between TIME and VOLUME in ICW and TBW. In rehydration test, the differences in the ICW and TBW among the four groups were compared, in which the differences were found between WS group 1 and NR group. Furthermore, the differences in the ICW and TBW between dehydration test and rehydration test in each group were analyzed. The contents of Table 2 were presented in the same order.
We have made revision accordingly in the Results Sections (Table 1: lines 276-295, Page 7; Table 2: lines 307-353, page 9).
Regarding Discussion Section, we summarized the results of the studies that were similar to the results of the current study or inconsistent with our study. Furthermore, we revised the descriptions of some studies and rearranged the order of the references (Lines 362-469, Pages 11-13).
As for the results of the supplementary data, we removed the results of the tables from the Results Section.
As for the BMI, we have revised accordingly in the Methods Section (Lines 212-213, Page 5).
